# Virus-Based Immunotherapy of Glioblastoma

**DOI:** 10.3390/cancers11020186

**Published:** 2019-02-05

**Authors:** Miika Martikainen, Magnus Essand

**Affiliations:** Department of Immunology, Genetics, and Pathology, Science for Life Laboratory, Uppsala University, 75185 Uppsala, Sweden; magnus.essand@igp.uu.se

**Keywords:** oncolytic virotherapy, cancer immunotherapy, glioblastoma

## Abstract

Glioblastoma (GBM) is the most common type of primary brain tumor in adults. Despite recent advances in cancer therapy, including the breakthrough of immunotherapy, the prognosis of GBM patients remains dismal. One of the new promising ways to therapeutically tackle the immunosuppressive GBM microenvironment is the use of engineered viruses that kill tumor cells via direct oncolysis and via stimulation of antitumor immune responses. In this review, we focus on recently published results of phase I/II clinical trials with different oncolytic viruses and the new interesting findings in preclinical models. From syngeneic preclinical GBM models, it seems evident that oncolytic virus-mediated destruction of GBM tissue coupled with strong adjuvant effect, provided by the robust stimulation of innate antiviral immune responses and adaptive anti-tumor T cell responses, can be harnessed as potent immunotherapy against GBM. Although clinical testing of oncolytic viruses against GBM is at an early stage, the promising results from these trials give hope for the effective treatment of GBM in the near future.

## 1. Current Status of Glioblastoma (GBM) Immunotherapy

Glioblastoma (GBM) is the most common and most aggressive primary malignant brain tumor in humans [1]. GBM can arise and be diagnosed either “de novo” or by evolving from less malignant astrocytomas or oligodendrogliomas [2]. The current standard care for GBM consists of radiotherapy, chemotherapy (temozolomide), and surgery when possible [3]. However, none of these interventions are curative, and GBM recurrence is virtually 100% certain. Consequently, the prognosis for GBM patients is dismal, with a median survival of only 14–15 months after diagnosis [3,4]. 

The recent success of cancer immunotherapy using immune checkpoint inhibitors (CPIs) against many otherwise untreatable cancers has raised expectations that such approaches could also be successfully used against GBM. For example, combination of anti-PD-1, anti-TIM-3, and targeted radiation showed impressive results in the preclinical mouse GL261 glioma model [5]. The drawback of using carcinogen-induced mouse models such as GL261 [6] is their relatively high immunogenicity and mutational load as compared with the clinical GBMs [7]. The results obtained in such preclinical models might thus overestimate the efficacy of immunotherapy against GBM. A recent study by Genoud et al. indicates that CPIs are not effective against the novel and significantly less immunogenic (and highly tumorigenic) SB28 model [8], which may better mimic the immune landscape of GBM. In line with this, experiences in the clinics show that GBM presents a particularly difficult target for immunotherapy, and the currently ongoing clinical trials utilizing CPIs, chimeric antigen receptor (CAR) T cells, or different vaccination strategies have resulted in mostly disappointing results [9,10,11,12,13,14,15]. Through whole-exome and RNA sequencing of a patient’s tumor and normal cells, neoantigens (derived from genetic mutations in tumor cells) can be predicted and evaluated as personalized vaccines. Two clinical trials have recently evaluated this approach for GBM and found that neoantigen-specific T cells can be generated and found inside GBM, showing that immunotherapy may be a fruitful way forward [16,17]. It is worth noting that PD-1 blockade has shown impressive results against hypermutated GBM [18,19], indicating that CPIs can be effective in a certain subset of patients. These cases can be associated with somatic mutations in the DNA mismatch-repair machinery [18,19]. Another glimmer of hope for effective GBM immunotherapy can be seen in the results of recent phase I/II clinical trials using cancer cell selectively replicating oncolytic viruses [20,21,22]. 

Oncolytic virotherapy is based on utilizing replicating viruses that can selectively kill the infected cancer cells. The virus-induced cell death can occur through a variety of different mechanisms, most (if not all) of which can be immunogenic (reviewed by [23]). During this process of immunogenic cell death (ICD), damage-associated molecular patterns (DAMPs) and tumor-associated antigens (TAAs), including patient-specific neoantigens caused by genomic mutations in cancer cells, are released from the disrupted cancer cells [24]. This in turn acts as a potent stimulus for the immune system and can, in the optimal case, lead into activation of effective antitumor immunity. In addition, the virus-induced antiviral innate immune response mediated by pathogen-associated molecular patterns (PAMPs) can act as a potent adjuvant to further boost antigen cross-presentation and consequent adaptive immune responses. Oncolytic viruses can thus be considered to function not only as directly cancer killing agents, but also as active anticancer vaccines [25,26]. Because of the recent appreciation of the immunostimulatory effects of oncolytic viruses, the focus of the field has clearly shifted from direct oncolysis to immunostimulatory properties of the viruses used. It is easy to hypothesize that virus-induced inflammation and ICD would be especially beneficial when treating heavily immunosuppressed (discussed later) tumors such as GBM. Importantly, use of oncolytic viruses together with radiation therapy or temozolomide has been shown to have synergistic activity [27,28,29], indicating that oncolytic virotherapy in combination with traditional forms of GBM therapy can be feasible.

Oncolytic viruses can also be used to transfer therapeutic payloads to the tumor. Viruses armed with immunoregulatory inserts such as interleukin 12 (IL-12) and OX40 ligand are currently being tested in clinical trials (Table 1). Other examples of clinically tested viruses with therapeutic payloads are gammaretrovirus “Toca 511” and vaccinia virus “TG6002”, which carry the cytosine deaminase (CD) gene (Table 1). When active in infected tumor cells, CD can convert the subsequently given 5-fluorocytosine drug into chemotherapeutic fluorouracil [30].

The oncolytic viruses used against GBM in current trials are listed in Table 1. Of these, perhaps the most promising viruses are DNX-2401, PVS-RIPO, and Toca 511, all of which have shown complete durable responses in approximately 20% of GBM patients who received virus intratumorally [20,21,22]. Virus-related severe adverse events in these trials have been rare. In Toca 511 and DNX-2401 trials, no dose-limiting toxicities were observed [21,22]. In the PVSRIPO trial, one (possibly virus-related) death and one dose-limiting toxic effect were reported [20]. Despite the early status of these trials, it is clear that oncolytic virotherapy is among the most compelling new therapies for GBM. The encouraging results obtained with PVS-RIPO, Toca511, and DNX-2401 have granted them a fast track designation by the U.S. Food and Drug Administration (FDA) for expedited drug review process. 

## 2. Heating the Suppressive Tumor Microenvironment with Viruses

The GBM microenvironment has proven to be a particularly challenging target for immunotherapeutic approaches. This can be largely attributed to the naturally isolated and tightly controlled brain microenvironment, which GBM thrives upon and skews even more immunosuppressive. The overall T cell infiltration in GBM is relatively poor and has a low CD8^+^/CD4^+^ T cell ratio [35]. Although cytotoxic CD8^+^ T cells can be found in patient GBM samples, they often display a PD-1^+^, LAG-3^+^, TIGIT^+^, CD39^+^, KLRG1^−^, and CD57^−^ profile, which is indicative of an exhausted phenotype of these cells [36]. GBM cells also attract microglia and macrophages that have been shown to promote immunosuppression and enhance tumor growth [37]. In fact, monocytes and macrophages can contribute to as much as 30% of the total amount of cells in gliomas [38], making them (in addition to the GBM cells themselves) a compelling target for therapy. In addition to the local immunosuppression, there is also evidence of GBM-mediated inhibition of peripheral immune responses. This is indicated by increased circulating regulatory T cells (Tregs) [39] and impaired cytokine response by peripheral blood lymphocytes [40]. It must also be noted that corticosteroids, which are used with standard GBM radiotherapy, chemotherapy (to alleviate swelling in the brain), also have notable immunosuppressive properties [41]. In preclinical studies, the corticosteroid dexamethasone has been shown to reduce serum neutralizing antibodies against oncolytic herpes simplex virus (HSV) G207 [42]. While having no effect on the direct oncolytic activity of the virus, the use of dexamethasone completely abolished virus-induced antitumor immunity against subcutaneous N18 neuroblastomas in A/J mice [42]. This indicates that temporal immunosuppression with corticosteroids could possibly be used early during oncolytic virotherapy to increase the efficacy of systemic virus delivery. The use of corticosteroids can, however, be clearly detrimental for the long-term antitumor immunotherapeutic effect.

Together with a low mutational burden and high heterogeneity, the immune evasive factors present in GBM (reviewed by [43]) are likely to render most immunotherapeutic approaches ineffective. There is, however, an increasing amount of evidence that oncolytic viruses can be used to disrupt a immunologically “cold” GBM microenvironment through induction of inflammation and ICD. One good example of ICD-driven oncolytic immunotherapy in an experimental mouse model is reported by Koks et al., using oncolytic Newcastle disease virus (NDV) in the syngeneic and fully immunocompetent GL261 model of orthotopic glioma [44]. Here, NDV was shown to induce ICD in GL261 cells, as detected by ectopic calreticulin surface expression, HMGB1 release, and increased cancer antigen expression. In line with the immunostimulatory potential seen in vitro, intratumoral administration of NDV was shown to lead to long-term survival, associated with elevated infiltration of cytotoxic T cells and reduced accumulation of myeloid-derived suppressor cells (MDSCs), in 50% of the treated mice. Similar evidence of an immunostimulatory therapeutic effect has been observed in the orthotopic GL261 model with adenovirus DNX-2401 (currently used in clinical trials). Here, intratumoral administration of DNX-2401 into GL261-OVA glioma-bearing mice enhanced the presentation of OVA epitopes to CD8^+^ T cells and had the potency to induce anti-glioma immunity [45]. Of note in the case of NDV, the long-term antitumor effect was reported to be lost in immunodeficient Rag2 knockout and in CD8-depleted mice, stressing the important role of functional immune system in the observed therapeutic effect [44]. Adding to the list of viruses that have shown survival benefit in mouse immunocompetent glioma models are the modified Semliki forest virus (SFV), VSV Δ51, and mouse-adapted Zika virus [46,47,48,49]. Interestingly, Zika virus has been reported to efficiently target and replicate in glioma stem cells [46], while viruses such as γ34.5-deleted HSV show restricted replication in this cell type [50]. Glioma stem cells, characterized as a highly self-renewing capable population of cells in the GBM tissue, have been proposed to be one of the main contributors to the high therapeutic resistance of GBM [51]. These cells, therefore, present a highly relevant therapeutic target, and the use of oncolytic viruses that are capable of efficiently infecting and killing these cells would be favorable.

The ability of oncolytic viruses to induce therapeutically relevant immune responses against GBM (at least in subsets of patients) has been demonstrated in clinical trials [20,21,52]. In fact, the immunostimulatory effect of oncolytic viruses used in the clinics can be speculated to be more relevant than the direct oncolytic power of the virus. Therapy with adenovirus DNX-2401 was shown to increase CD8^+^, T-bet^+^ T cell infiltration together with a reduction of TIM-3 expression [21]. Parvovirus H-1 therapy was noted to activate GBM-associated microglia (Cathepsin B expression) and, similar to DNX-2401 therapy, increase cytotoxic T cell infiltration [52]. In addition, PVS-RIPO therapy has been reported to induce notable GBM tissue inflammation, as evidenced by the “soap bubble” appearance in magnetic resonance imaging [20]. One classic sign of immunotherapy-induced inflammation is also an apparent initial growth of the tumor. This so-called pseudoprogression is caused by increased immune cell infiltration to the tumor, not by the growth of tumor cells [53]. Pseudoprogression has been observed, for example, after PVS-RIPO infusions as increased cerebral edema. Although this type of tumor swelling is therapeutically favorable, it proposes challenges for the interpretation of radiologic images and patient care.

Dendritic cells (DCs) are professional antigen presenting cells that are essential in the induction of adaptive immunity. Owing to their capability to activate T cells, DCs function in the interface between the innate and adaptive immune system. DC activation has been shown to have an especially potent adjuvant effect in the promotion and development of antitumor immunity against GBM. For example, DC stimulation with Toll-like receptor 3 (TLR3) agonists enhances the anti-tumor immune response to anti-PD-1 therapy in the orthotopic GL261 glioma model [54]. A potent adjuvant effect has also been observed in the subcutaneous B16 melanoma model using heat-inactivated vaccinia virus, which can activate the cytosolic DNA-sensing cGAS-STING-pathway in DCs [55]. It is likely that the improved therapeutic activity of inactivated virus in this case can be explained by the profound immunosuppressive effect of fully replicative vaccinia. This study also points out that Batf3-dependent CD103^+^/CD8α^+^ DCs, also known as type-1 conventional DCs, which excel at cross presentation of TAAs to CD8^+^ T cells [56], are crucial for the observed therapeutic effect. 

Interestingly, PVS-RIPO was also recently reported to be able to directly infect dendritic cells and macrophages [57]. Although PVS-RIPO propagation in these cells was shown to be non-lethal and only marginally productive, it resulted in a robust IFN-I response (shown by STAT1 Y701 phosphorylation and expression of interferon-stimulated genes IFIT1, ISG15, and PD-L1), accompanied by the expression of costimulatory molecules and cytokines [57]. It is, however, not clear how big a role this effect has in the therapeutic effects seen in patients treated with PVS-RIPO.

Taken together, the increasing amount of evidence suggests that oncolytic viruses can be used as potent immunotherapy against GBM. Given the immunologically cold GBM microenvironment, it is likely that potent “de novo” induction of antitumor T cell immunity is needed for effective therapy. In the context, oncolytic virotherapy can be seen to have an especially favorable effect, both disrupting the system by destruction of cancer cells and re-stimulating both the innate and adaptive immunity against the infected cancer cells.

## 3. The Prospect of Combining Oncolytic Viruses with Checkpoint Inhibitors in GBM Therapy

During recent years, the use of immune checkpoint inhibitors (CPIs) has shown great success in inducing long-term complete remissions in patients with otherwise refractory cancers [58]. Since 2011, when the first immune CPI was approved by the FDA for treatment of melanoma, these novel drugs have been rapidly approved for many different types of cancers including melanoma, head and neck squamous cell carcinoma, non-small cell lung cancer, renal cell carcinoma, Hodgkin’s lymphoma, and urothelial carcinoma [58]. The use of CPIs can also result in therapeutic response against primary or metastatic brain tumors [18,19,59]. It is unclear to what extent the observed responses are the result of CPIs’ ability to get across the blood–brain barrier, as peripheral immune activation might be effective enough to induce antitumor immunity. There are currently several ongoing clinical trials of CPIs in glioblastoma, including phase III trials with Ipilimumab (blocking CTLA-4) and Nivolumab (blocking PD-1) (NCT02017717, NCT02617589).

CPIs work by blocking the negative regulators of T cell function, thereby sustaining T cell activity. Consequently, tumors with low T cell infiltration (such as GBM) are unlikely to get significant benefit from CPI therapy. In these cases, it would be important to induce T cell infiltration into the tumor prior to CPI therapy. As oncolytic viruses have a potent ability to induce CD8^+^ T cell responses against tumor cells, it can be expected that precursory oncolytic virotherapy would enhance the effectiveness of CPIs. The combination of checkpoint blockade with oncolytic virotherapy is also an attractive option because virus-induced inflammatory response in the tumor can lead to upregulation of PD1 on T cells and PD-L1 on tumor cells [60]. Taken together, there is a clear rationale for combining CPIs with precursory therapy with oncolytic viruses for synergistic and effective immunotherapy of immunologically cold tumors, such as GBM.

Many different viruses have been tested together with CPIs in different preclinical tumor models with encouraging results [61]. For example, intratumoral injection of Newcastle disease virus (NDV) into B16 melanomas has been shown to induce CD4^+^ and CD8^+^ T cell infiltration in both infected and distant (non-infected) tumors, rendering them vulnerable to anti-CTLA-4 blockade [62]. As another good example of synergy between oncolytic viruses and CPIs, recombinant adenovirus (hTertAd) infection/oncolysis combined with anti PD-1 therapy was shown to broaden the spectrum of tumor neoantigen-specific T cell response in the subcutaneous CMT64 lung adenocarcinoma model [63]. A notable added benefit of combining oncolytic virotherapy and checkpoint blockade has also been observed in immunocompetent mouse glioma models. For example, measles virus together with anti-PD-1 [64], VSV (expressing TAAs HIF-2α, Sox-10, and c-Myc) together with anti-PD-1 [65], adenovirus Delta-24-RGDOX (expressing the immune costimulatory OX40 ligand) combined with anti-PD-L1 [66], and reovirus combined with anti-PD-1 antibodies [60] have shown therapeutic benefits against the GL261 glioma model. In addition, HSV (G47Δ-mIL12) together with anti-PD-1 and anti-CTLA-4 has shown therapeutic benefits in the mouse 005 glioma stem cell model [67]. These two immunotherapies have also been combined by engineering oncolytic herpes simplex virus, which expresses single-chain antibody fragments (scFv) against PD-1 [68]. According to the results by Passaro et al., GBM cells infected with this virus express and secrete scFvPD-1, which can bind to mouse PD-1 without affecting the oncolysis. It remains to be seen whether this design of oncolytic viral constructs can also be functional under clinical settings. 

Because of the success of checkpoint inhibitors against other types of tumors and the evident synergy with oncolytic viruses, it is not a big surprise that many virus/antibody combinations are currently investigated in clinical trials [61,69]. Of note, a phase 2 clinical trial with DNX-2401 + pembrolizumab (anti-PD-1 antibody) is currently recruiting patients with recurrent GBM or gliosarcoma (NCT02798406, Table 1).

## 4. Feasibility of Systemic Virus Delivery into the GBM

Because of the blood–brain barrier, the brain is a notoriously difficult target for efficient drug delivery. GBM are characterized by highly abnormal tumor vessels, but although GBM vasculature can be considered leaky, regions with an intact blood–brain barrier are also present (reviewed by [70]) and can limit effective systemic delivery of drugs into the tumor. In most of the clinical trials, the virus is administered intratumorally or into the resection cavity. The clear rationale for this strategy is to ensure efficient direct tumor cell infection. Systemic delivery of oncolytic virus, if done safely, could, however, present a less technically challenging option. Systemic delivery would also be more broadly applicable in disseminated cases or if the tumor cannot be reached directly. A peripheral administration route could be used to stimulate therapeutically beneficial systemic immune responses. 

Some viruses (such as SFV) have natural tropism to the central nervous system (CNS), which can be harnessed to achieve potent delivery of virus into the brain. At least, SFV [47,48,71], vaccinia virus [72,73], chimeric vesicular stomatitis virus (VSV) [74], parvovirus H-1 [75], and picornavirus SVV-001 [76] have shown the ability to infect intracranial tumors in animal models when administered systemically. Notably, parvovirus H-1 and reovirus have been shown to be able to reach GBM tumors following intravenous injection in clinical trials [32,52,60], indicating that effective systemic delivery of the virus can also be achieved in human patients. 

Classically, induction of virus neutralizing antibodies (NAbs) is considered to limit repeated virus injections or the use of virus strains that the patients’ immune system has previously encountered. Interestingly, studies by Berkeley et al. challenge this paradigm by showing that monocytes can uptake and internalize reovirus–NAb complexes (mediated by FcγRIII) and release functional replication competent reovirus to tumor [77]. As GBM is naturally heavily infiltrated by peripheral monocytoid cells, this “trojan horse” strategy might present an interesting (if possibly limited to certain viruses/strains) new way of efficient systemic virus delivery.

## 5. Innate Antiviral Response as a Challenge and Opportunity for Virus-Based Therapy of GBM

The first line of defense against viral infections is innate antiviral response, which is largely orchestrated by type-I interferons (IFN-I). IFN-Is are secreted by infected cells in response to PAMPs (e.g., viral nucleic acids) detected by host cell pattern recognition receptors (PRRs). Generally speaking, the IFN-I response plays a major role in inducing an antiviral state (via signaling through the IFNAR–JAK–STAT pathway) in the surrounding cells, thereby limiting viral spread. In addition, IFN-Is promote antigen presentation, natural killer (NK) cell function, and the development of antigen-specific adaptive immune responses [78]. In the GBM context, however, the effect of IFN-Is seems to be more contradictory or unclear. On the one hand, studies in mouse models show that IFNAR1-deficiency accelerates gliomagenesis by impairing immunosurveillance (seen as increased infiltration of CD11b^+^/Ly6G^+^ and CD4^+^/FoxP3^+^ cells, but decreased Tc1 effector cells and CD11c^+^ DCs) [79]. Also supporting the antitumor effect of IFN-I signaling, high expression of negative IFN-I regulatory factor ATM (ataxia-telangiectasia mutated kinase) in GBM has been shown to correlate with poor patient survival [80]. On the other hand, the constitutively activated IFN-I signaling pathway [81] and constitutive activation of STAT proteins have been observed in GBM [82]. Autocrine IFN-I signaling has also been suggested to contribute to the immune evasion of glioma cells [81], probably by negatively regulating the antigen-presenting capacity of glial cells [83]. Interestingly, chronically elevated levels of IFN-I in the brain have been shown to be related to ageing [83]. As GBM mostly affects people at an older age (median age of diagnosis is 64) [4], it would then also be possible that these tumors are evolved to tolerate or even to employ active IFN-I signaling to their benefit.

Many of the currently used oncolytic viruses are either natural or the result of genetic engineering sensitive to the antiviral effect of IFN-Is. The rationale for use of such viruses is that cancer cells in many cases have dysfunctional IFN-I signaling [84]. Notably, human GBM samples have been reported to both respond to and produce IFN-I [85,86,87]. This, together with possibly abnormal effects of IFN-Is on GBM immunosurveillance (described above), would indicate that, although considered an “Achilles heel” of cancer cells, a defective IFN-I system cannot be taken for granted in GBM. Thereby, viruses that have at least some resistance to innate antiviral signaling would be favorable. Indeed, viruses such as myxomavirus, certain versions of SFV, and clinically tested poliovirus PVS-RIPO show resistance/insensitivity to IFN-I mediated antiviral signaling [48,88,89]. Importantly, SFV and PVS-RIPO do, however, activate IFN-I response in the infected cells. The use of IFN-I-tolerant viruses raises safety issues by possibly enhancing the risk of an uncontrolled spread of the virus in healthy cells. In the case of PVS-RIPO, unwanted replication in neurons is inhibited by replacing the poliovirus internal ribosomal entry site (IRES) with the IRES from human rhinovirus type 2, which is not functional in normal neuronal cells [90]. Neuronal replication of SFV (another (+)ssRNA virus, studied for its oncolytic properties by us and others) can be controlled by microRNA-mediated detargeting [47,48,91]. 

The virus infection-induced release of IFN-I and other DAMPs attracts innate immune cells to eat and destroy the infected cells. While this is an important step in priming adaptive antitumor (and antiviral) immunity, the premature killing of infected cells can also lead to suboptimal virus spread in the tumor. For example, HSV therapy has been shown to increase infiltration of macrophages and microglia in the mouse GBM model [92]. Although shown to be polarized toward favorable proinflammatory M1 phenotype (with high levels of CD86, MHCII, and Ly6C), increased tumor necrosis factor alpha (TNFα) produced by these cells induced apoptosis in infected tumor cells and ultimately inhibited viral spread. A negative effect of the innate immune system attacking the HSV-infected GBM cell in terms of replication has been shown in another study, in which temporal NK-cell depletion by administration of transforming growth factor beta (TGF-β) improved therapeutic effect of the virus [93]. NK cells and infiltrating monocytes/macrophages have also been implicated to play an important role in the early clearance of oncolytic Myxoma virus and HSV in syngeneic glioma models [94,95]. Notably, tumor-associated microglia (and astrocytes) have been shown to attenuate the replication of the oncolytic vaccinia virus LIVP 1.1.1 in murine GL261 gliomas by acting as vaccinia virus traps [96]. 

On the basis of the findings in preclinical GBM models, temporal inhibition of innate responses in the early stage of viral therapy could lead to increased viral spread in the GBM microenvironment. This would in turn lead to more potent oncolysis and subsequent immunotherapeutic effect. Alternatively, viruses that can tolerate and escape the innate antiviral effect could be used (if safety can be assured). It is likely that preclinical GBM models do not perfectly recapitulate the innate immune microenvironment of human GBM, making direct translation of results difficult. In addition, human GBMs show high heterogeneity both within the same tumors and between different patients [97]. Because of these limitations, it is reasonable to hypothesize that viruses that potently activate the innate immune system (inducing robust IFN-I response) and resist the following antiviral response (without risk of unwanted spread in healthy tissues) would have the best chance of replicating in GBM and, consequently, the best chance of inducing curative therapeutic response. 

## 6. Conclusions

The use of viruses as oncolytic agents is not a particularly new idea. While the focus used to be on the direct oncolytic effect, the current working model of oncolytic virotherapy rather relies on utilizing viruses as immunotherapeutic agents. As oncolytic viruses can replicate in the target cells, they provide an intriguing possibility for one shot self-amplifying cancer therapy. On the other hand, it is also likely that repeated dosing could be beneficial (if not crucial) for the therapeutic effect through immunostimulatory effect as therapeutic vaccines. Intratumoral administration clearly has the benefit of delivering the virus directly to the tumor, thereby mostly evading virus neutralization by systemic antiviral antibodies, while the systemically delivered virus has the potential to break through the blood–brain barrier and reach tumor cells and sites not reachable by intratumoral injections. The effect on antitumor immune response from stimulating systemic antiviral immune response must be further studied before conclusions can be made. It is possible that combinations of intratumoral and intravenous doses of oncolytic viruses would produce synergistic therapeutic benefits by inducing both local and systemic immune responses. Direct side-by-side comparison of immune-related characteristics of the different viruses being developed and used in preclinical and clinical settings would also be valuable in order to evaluate the best candidates for GBM therapy. Nevertheless, recent phase I/II clinical trials demonstrate that the use of oncolytic viruses can significantly improve the survival of some GBM patients [20,21,22]. Most of the therapeutic benefits so far in the trials can be associated with the potent stimulation of antitumor immune responses, thereby warming the naturally cold GBM immune microenvironment (Figure 1A). Later phase clinical trials will demonstrate whether therapy with oncolytic viruses such as PVS-RIPO, DNX-2401, and Toca511 (currently on fast track status by the FDA) can lead to truly curative GBM therapy.

The addition of therapeutic payloads to oncolytic viruses to boost antitumor potency is an attractive option. The increased therapeutic effect of such an armed virus construct is naturally dependent on the efficient replication of the virus in the target cells. Robust replication-induced cell death can, however, lower the amount of transgene produced by the infected cell. However, adding genes or replacing viral genes with therapeutic ones can hamper viral replication potency. Therefore, any oncolytic virus construct that depends on delivery and expression of therapeutic genes must be carefully optimized. It must also be considered that excessive expression of immunoregulatory transgenes might have unwanted toxic side-effects [98]. 

The GBM microenvironment presents a challenge for oncolytic virus delivery into the tumor, as well as effective viral replication and spread within the tumor (Figure 1B). Although the functionality and exact role of the innate immune system in GBM is still elusive, it is reasonable to speculate that at least some replicative power, despite antiviral signaling, would be beneficial for viral spread in the tumor. In addition, the heavy infiltration of innate immune cells (in both human and mouse models) and the presence glioma stem cells can render the GBM tissue resistant to effective viral spread, pointing out the need for robustly replicating viruses in order to disrupt the suppressive GBM microenvironment. In optimal cases, oncolytic viruses can serve as versatile innate adjuvants for boosting adaptive anti-tumor immunity. Notable overlap between virotherapy and other forms of immunotherapy, together with the possibility of turning an immunosuppressive tumor microenvironment into an immune-vulnerable one, opens interesting possibilities for combination therapies. It is likely that synergistic combination therapy with, for example, immune checkpoint inhibitors will ultimately be needed for a curative GBM therapy. 

More work is needed to fully unlock the potential of viruses as effective anticancer agents. However, virus-based immunotherapy clearly shows promise in leading to curative treatment of currently incurable tumors such as GBM in the near future. 

## Figures and Tables

**Figure 1 cancers-11-00186-f001:**
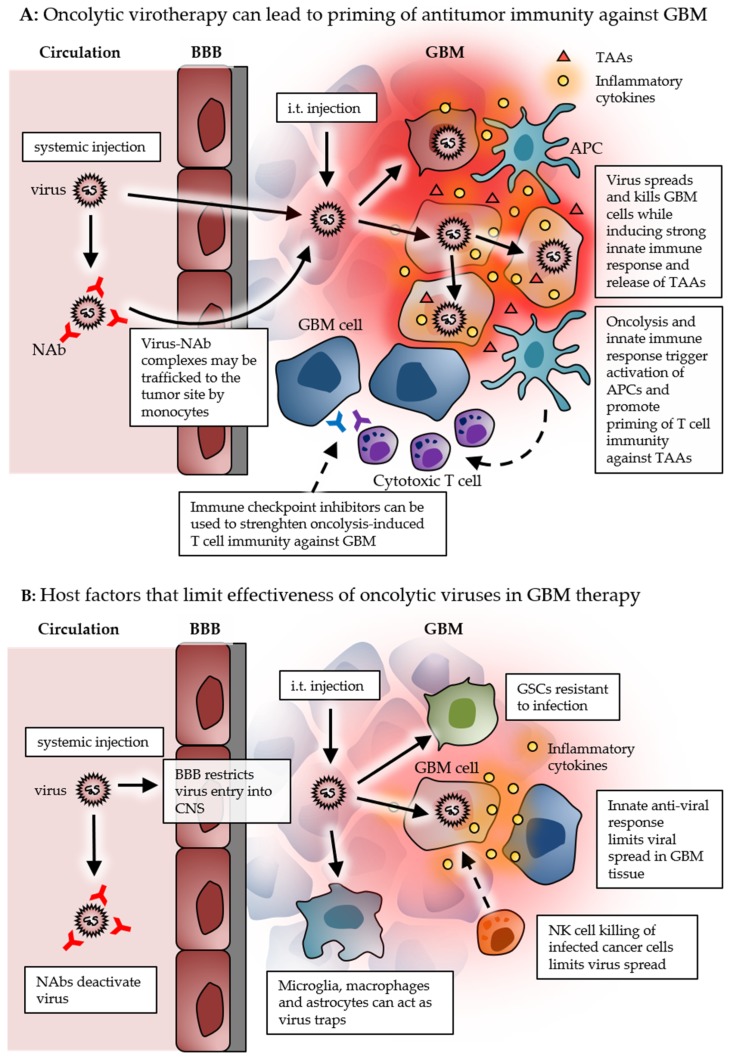
Schematic picture summarizing the events and factors related to successful oncolytic immunotherapy (**A**), and to limited effectiveness of oncolytic viruses (**B**) in GBM. Abbreviations: BBB: blood–brain barrier, GBM: glioblastoma, TAA: tumor-associated antigen, NAb: neutralizing antibody, APC: antigen-presenting cell, CNS: central nervous system, GSC: glioma stem cell, NK: natural killer cell.

**Table 1 cancers-11-00186-t001:** Currently active or completed clinical trials utilizing replication-competent viruses against malignant gliomas. Search results from clinicaltrials.gov.

**Currently Active or Recruiting**					
**Adenovirus**	**Virus Construct**	**Phase**	**Therapy Regimen**	**Trial No.**	**Results**
DNX-2401 + pembrolizumab	Deletion in E1A. RGD-4C fiber modification	II	IT injection followed by Pembrolizumab (IV) every 3 weeks	NCT02798406	-
DNX-2440	DNX-2401 (above) armed with OX40L	I	IT injection	NCT03714334	-
CRad-S-pk7 (loaded into neural stem cells)	E1A under survivin promoter, pK7 fiber modification	I	injection into resection cavity	NCT03072134	-
**HSV**	**Virus Construct**	**Phase**	**Therapy Regimen**	**Trial No.**	**Results**
C134	γ34.5 deletion IRS1 under HCMV promoter	I	IT injection	NCT03657576	-
M032	γ34.5 deletion Armed with human IL-12	I	single IT infusion	NCT02062827NSC733972	-
rQNestin 34.5	One copy of γ34.5 under nestin promoter With UL39 deletion	I	IT with/without preceding IV cyclophosphamide	NCT03152318	-
G207	γ34.5 deletion Inactivating insertion of lacZ in UL39 gene	I	IT infusion	NCT02457845	-
**Other**	**Virus Construct**	**Phase**	**Therapy Regimen**	**Trial No.**	**Results**
Vaccinia virus TG6002 + 5-FC	TK and RR deletion Expresses cytosine deaminase	I/II	3 weekly IV infusions, followed by oral 5-FC	NCT03294486	-
Measles Virus (MV-CEA)	expresses CEA	I	injection into resection cavity and/or IT	NCT00390299	-
Poliovirus (PVSRIPO)	attenuated (Sabin) poliovirus with IRES from HRV2	I/Ib	IT via convection- enhanced delivery	NCT01491893NCT03043391	[20]
Reovirus (REOLYSIN) + sargramostim (rGM-CSF)	unmodified reovirus	I	repeated cycles of sargamostim followed by IV virus	NCT02444546	-
Toca 511 + Toca FC	described above	II/III	injection into resection cavity followed by oral Toca FC	NCT02414165	-
**Completed**					
**Adenovirus**	**Virus Construct**	**Phase**	**Therapy Regimen**	**Trial No.**	**Results**
DNX-2401 + Temozolomide	described above	I	injection in the brain parenchyma followed by temozolomide	NCT01956734	-
DNX-2401 + IFNγ	described above	I	IT injection followed by IFNγ	NCT02197169	-
DNX-2401	described above	I/II	intracerebral infusion	NCT01582516	[21]
**Other**	**Virus Construct**	**Phase**	**Therapy Regimen**	**Trial No.**	**Results**
HSV G207	described above	I/II	IT injection	NCT00028158	[31]
Parvovirus H-1PV	unmodified rat parvovirus	I/II	combinations of IV, IT, and resection cavity injections	NCT01301430	[32]
Reovirus (REOLYSIN)	unmodified human reovirus	I	IT infusion	NCT00528684	[33]
Toca 511 + Toca FC	described above	I	injection in resection cavity followed by oral Toca FC	NCT01470794	[22,34]

Abbreviations: IT: intratumoral, IV: intravenous, HSV: herpes simplex virus, HCMV: human cytomegalovirus, IL-12: interleukin 12, TK: thymidine kinase gene, RR: ribonucleotide reductase gene, 5-FC: 5-fluorocytosine, CEA: carcinoembryonic antigen, IRES: internal ribosomal entry site, HRV2: human rhinovirus 2, rGM-CSF: recombinant granulocyte-macrophage colony-stimulating factor, IFN-γ: interferon gamma.

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
