# Peer review of "Virus-Based Immunotherapy of Glioblastoma"

_cancers, 2019, doi:10.3390/cancers11020186_

Round 1
Reviewer 1 Report
Overall a nice general review of the field of viral therapy and immunotherapy by Drs. Martikainen and Essand. Some important points which could be added:
1) Some patients with glioblastoma (mostly in cases of MMR deficiency) have responded to CPI, suggesting that CPI in the right context can be effective
2) Table of differences/similarities between viruses would be helpful. Including key immune-related characteristics such as some viruses being IFN-I tolerant (mentioned on lines 240-241).
3) Briefly mention safety/toxicity in trials.
4) For clinical trials in table 1, especially completed trials, please provide the observation in the table if possible. For example median survival times, %SD,PRs etc…
5) The potential use of viral payloads
Other comments:
Line 33- Also mentioned a recent paper by Paul Walker showing that in SB28 model, unlike GL261, anti-PD1 was not effective in a poorly immunogenic mouse model. Also please mention the weaknesses of using a carcinogen-induced model such as GL261 for research.
For citation (6)- Please cite primary articles and be a bit more specific about results of CAR-T cell trials.
Sentence on line 42-43 on DAMPs should have citations
Lines 85-86 on "cold" GBM should have a citation
Line 89-define CRT
Lines 103-108 on GSCs does not seem to fit in this section
Line 110 needs citations
Line 117-118... for pseudoprogression, especially immunotherapy-induced pseudoprogression this may be due to increased immune infiltrate at the tumor site. Please consider mentioning this. Also please make it clear on line 117 that pseudoprogression is not from the tumor cell growth.
Line 139 "very cold" may be an overstatement. Consider alternative word
Line 155 should have a citation. Also, it should be noted that some tumors have T-cells at the margin which enter the tumor upon CPI therapy ..work of Jerome Galon.
Line 193- please specify which tumors the author has in mind that cannot be reached directly to inject the virus?
Line 275- “can significantly improve the survival of GBM patients”. Make it clear that only some patients respond
Author Response
Overall a nice general review of the field of viral therapy and immunotherapy by Drs. Martikainen and Essand. Some important points which could be added:
1) Some patients with glioblastoma (mostly in cases of MMR deficiency) have responded to CPI, suggesting that CPI in the right context can be effective
This is true. We have added the following text to point this (Lines 45-48)
“Of notice, PD-1 blockade has shown impressive results against hypermutated GBM [18,19], indicating that CPIs can be effective in certain subset of patients. These cases can be associated to somatic mutations in the DNA mismatch-repair machinery [18.19].”
2) Table of differences/similarities between viruses would be helpful. Including key immune-related characteristics such as some viruses being IFN-I tolerant (mentioned on lines 240-241).
The reviewer raises a very important point. However, there is no direct side-by-side comparison of IFN sensitivity (or other immune-related characteristics) of different oncolytic viruses available. We therefore feel that pointing out the differences/similarities using currently available results is difficult and possibly misleading.
To stress this point, we have added the following text (lines 318-321)
“Direct side-by-side comparison of immune-related characteristics of the different viruses being developed and used in preclinical and clinical settings would also be valuable in order to evaluate the best candidates for GBM therapy.”
3) Briefly mention safety/toxicity in trials.
We have added the following text (lines 78-80)
“Virus-related severe adverse events in these trials have been rare. In Toca 511 and DNX-2401 trials, no dose limiting toxicities were observed [21,22]. In the PVSRIPO trial, one (possibly virus-related) death and one dose-limiting toxic effect were reported [20].”
4) For clinical trials in table 1, especially completed trials, please provide the observation in the table if possible. For example median survival times, %SD,PRs etc…
We have added references to published results (if available) into the table.
5) The potential use of viral payloads
We have added the following paragraphs (lines 69-74):
“Oncolytic viruses can also be used to transfer therapeutic payloads to the tumor. Viruses armed with immunoregulatory inserts such as IL-12 and OX40 ligand are currently tested in clinical trials (Table 1.). Other examples of clinically tested viruses with therapeutic payloads are gammartrovirus “Toca 511” and vaccinia virus “TG6002” which carry cytosine deaminase (CD) gene (Table 1.). When active in infected tumor cells, CD can convert subsequently given 5-fluorocytosine drug into chemotherapeutic fluorouracil [30]
And lines 328-335
“The addition of therapeutic payloads to oncolytic viruses to boost antitumor potency is an attractive option. The increased therapeutic effect of such armed virus construct is naturally dependent on efficient replication of the virus in the target cells. Robust replication-induced cell death can however lower the amount of transgene produced by the infected cell. On the other hand, adding genes or replacing viral genes with therapeutic ones can hamper viral replication potency. Therefore, any oncolytic virus construct that depends on delivery and expression of therapeutic genes must be carefully optimized. It must also be considered that excessive expression of immunoregulatory transgenes might have unwanted toxic side-effects [42]. “
Other comments:
Line 33- Also mentioned a recent paper by Paul Walker showing that in SB28 model, unlike GL261, anti-PD1 was not effective in a poorly immunogenic mouse model. Also please mention the weaknesses of using a carcinogen-induced model such as GL261 for research.
We have added the following text (lines 31-38):
“For example, combination of anti-PD-1, anti-TIM-3 and targeted radiation showed impressive results in the preclinical mouse GL261 glioma model [5]. The drawback of using carcinogen-induced mouse models such as GL261 [6] is their relatively high immunogenicity and mutational load as compared the clinical GBMs [7]. Results obtained in such preclinical models might therefore overestimate the efficacy of immunotherapy against GBM. A recent study by Genoud et al. indicates that CPIs are not effective against the novel and significantly less immunogenic (and highly tumorigenic) SB28 model [8], which may better mimic the immune landscape of GBM.”
For citation (6)- Please cite primary articles and be a bit more specific about results of CAR-T cell trials.
We have added citations [9-15] to original articles. As CAR T-cell therapy is not in the focus of our review, we wish not to go into details of these trials.
Sentence on line 42-43 on DAMPs should have citations
We have added citation [24] (line 56)
Lines 85-86 on "cold" GBM should have a citation
We have changed the text as follows:
”There is however increasing amount of evidence that oncolytic viruses can be used to disrupt immunologically “cold” GBM microenvironment through induction of inflammation and ICD (as discussed below)”
Line 89-define CRT
We have replaced the abbreviation CRT with calreticulin (line 117)
Lines 103-108 on GSCs does not seem to fit in this section
We have modified the text to better fit the section (lines: 129-135)
Line 110 needs citations
We have added the citations [20,21,52] (line 137)
Line 117-118... for pseudoprogression, especially immunotherapy-induced pseudoprogression this may be due to increased immune infiltrate at the tumor site. Please consider mentioning this. Also please make it clear on line 117 that pseudoprogression is not from the tumor cell growth.
We have changed the text as follows (lines 144-146):
“One classic sign of immunotherapy-induced inflammation is also an apparent initial growth of the tumor. This so called pseudoprogression is caused by increased immune cell infiltration to the tumor, not by the growth of tumor cells [53].”
Line 139 "very cold" may be an overstatement. Consider alternative word
We have changed the text from “very cold” to “cold” (line 169)
Line 155 should have a citation. Also, it should be noted that some tumors have T-cells at the margin which enter the tumor upon CPI therapy ...work of Jerome Galon.
Although this can be true, we could not find a good reference for this and chose not include it into the text.
Line 193- please specify which tumors the author has in mind that cannot be reached directly to inject the virus?
We do not refer to any specific type of tumor or any specific tumor location. We have re-written the text as follows (line 228):
“Systemic delivery would also be more broadly applicable in disseminated cases or if tumor cannot be reached directly”
Line 275- “can significantly improve the survival of GBM patients”. Make it clear that only some patients respond
We have changed the text to “can significantly improve the survival of some GBM patients” (line 322)
Reviewer 2 Report
This is a timely review of an important field in oncolytic virus therapy.
The authors present a concise overview of preclinical and clinical oncolytic virus research in GBM.
The manuscript is generally well written and accurate. Whilst a good range of topics were discussed, the following additions would improve the impact of the manuscript:
I would like to see more detail in Table 1 to include a column on virus structure/engineering, as well as a brief description of each trial.
I would like to see more discussion of the advantages and disadvantages of intratumoral versus IV injection of virus, for the treatment of GBM, keeping in mind the likelihood of virus reaching tumour, the generation of NABs and limits to the potential administrable dose.
I would also like to see some discussion on single versus repeat dosing, bearing in mind that this is difficult to achieve via intratumoural injection.
The authors mention combination with checkpoint blockade, but do not address the difficulty in delivery of antibody to GBM across the BBB. Does this matter, e.g. does peripheral engagement of Ab with immune cells, which then themselves traffic to GBM circumvent this problem? Should we be concentrating on virus-encoded antibodies instead?
Many GBM patients take steroids, is this necessarily bad for virus efficacy?
Author Response
This is a timely review of an important field in oncolytic virus therapy.
The authors present a concise overview of preclinical and clinical oncolytic virus research in GBM.
The manuscript is generally well written and accurate. Whilst a good range of topics were discussed, the following additions would improve the impact of the manuscript:
I would like to see more detail in Table 1 to include a column on virus structure/engineering, as well as a brief description of each trial.
We have added more details to the table including brief descriptions of the virus constructs and the therapy regimen.
I would like to see more discussion of the advantages and disadvantages of intratumoral versus IV injection of virus, for the treatment of GBM, keeping in mind the likelihood of virus reaching tumour, the generation of NABs and limits to the potential administrable dose.
I would also like to see some discussion on single versus repeat dosing, bearing in mind that this is difficult to achieve via intratumoural injection.
To address these points, we have added the following text to the summary part (lines 310-321):
As oncolytic viruses can replicate in the target cells, they provide an intriguing possibility for one shot self-amplifying cancer therapy. On the other hand, it is also likely that repeated dosing could be beneficial (if not crucial) for the therapeutic effect through immunostimulatory effect. Intratumoral administration clearly has the benefit of delivering virus directly to tumor thereby mostly evading virus neutralization by systemic antiviral antibodies. The possible negative or positive effects of stimulating systemic antiviral immune response on the antitumor response must however be further studied before conclusion can be made. It is possible that combinations of intratumoral and intravenous doses of oncolytic viruses would produce synergistic therapeutic benefit by inducing both local and systemic immune responses. Direct side-by-side comparison of immune-related characteristics of the different viruses being developed and used in preclinical and clinical settings would also be valuable in order to evaluate the best candidates for GBM therapy.
The authors mention combination with checkpoint blockade, but do not address the difficulty in delivery of antibody to GBM across the BBB. Does this matter, e.g. does peripheral engagement of Ab with immune cells, which then themselves traffic to GBM circumvent this problem? Should we be concentrating on virus-encoded antibodies instead?
We have added the following text (lines 180-183):
”The use of CPIs can also result in therapeutic response against primary or metastatic brain tumors [18,19,59]. It is unclear to what extend the observed responses are due to CPIs ability to get across the blood brain barrier as peripheral immune activation might be effective enough to induce antitumor immunity.”
Many GBM patients take steroids, is this necessarily bad for virus efficacy?
We have added the following text (lines 102-109):
“In preclinical studies, the corticosteroid dexamethasone has been shown to reduce serum neutralizing antibodies against oncolytic HSV G207 [42]. While having no effect on the direct oncolytic activity of the virus, the use of dexamethasone completely abolished virus-induced antitumor immunity against subcutaneous N18 neuroblastomas in A/J mice [42]. This indicates that temporal immunosuppression with corticosteroids could possibly be used early during oncolytic virotherapy to increase efficacy of systemic virus delivery. The use of corticosteroids can however be clearly detrimental for the long-term therapeutic effect.”